# Foraging Behaviors of Red Imported Fire Ants (Hymenoptera Formicidae) in Response to Bait Containing Different Concentrations of Fipronil, Abamectin, or Indoxacarb

**DOI:** 10.3390/insects14110852

**Published:** 2023-10-31

**Authors:** Chengju Du, Hailong Lyu, Lanfeng Wang, Lei Mao, Lin Li, Xinya Yang, Cai Wang

**Affiliations:** 1College of Forestry and Landscape Architecture, South China Agricultural University, Guangzhou 510642, China; duchengju@stu.scau.edu.cn (C.D.);; 2Guangzhou Guangjian Construction Engineering Testing Center Co., Ltd., Guangzhou 510699, China

**Keywords:** *Solenopsis invicta*, bait, fipronil, foraging behavior, recruitment, repellency

## Abstract

**Simple Summary:**

Both field and laboratory studies showed that 0.0125% fipronil bait is repellent against *S. invicta* workers; therefore, higher concentrations of fipronil should be avoided in fire ant bait production. In future studies evaluating the effectiveness of fire ant baits, we suggest considering the effect of active ingredients and their concentrations on bait acceptance.

**Abstract:**

The red imported fire ant, *Solenopsis invicta* Buren, is a severe pest with agricultural, ecological, and medical significance. The baiting treatment is one of the main methods to control *S. invicta*. However, few studies have evaluated the acceptance of fire ant bait. Here, field and laboratory studies were conducted to investigate the foraging behaviors of *S. invicta* responding to fire ant baits containing different concentrations of active ingredients (fipronil, abamectin, or indoxacarb). Field studies showed that *S. invicta* transported significantly less 0.0125% fipronil bait than control bait (without toxicant) and 0.0001% fipronil bait. The number of foraging ants significantly decreased with an increase in fipronil concentration. Our previous study showed that *S. invicta* usually buries the food treated with repellent chemicals, and interestingly, significantly more soil particles were transported into tubes containing 0.0001% fipronil bait than tubes containing control bait or 0.0125% fipronil bait. In addition, *S. invicta* transported significantly less 0.0005% abamectin bait than control bait, and significantly fewer ants were found in tubes containing 0.0125% abamectin bait than control bait. However, there was no significant difference in bait transport, number of foraging ants, and weight of soil particles relocated in tubes containing different concentrations of indoxacarb bait. In addition, laboratory studies showed that *S. invicta* transported significantly less 0.0125% fipronil bait than control bait and bait containing abamectin (0.0025% or 0.0125%) or indoxacarb (0.0125% or 0.0625%). In addition, the transport speed for the 0.0125% fipronil bait was the slowest. These results show that specific concentrations of some active ingredients may negatively affect bait acceptance for *S. invicta*, and should be avoided in fire ant bait production.

## 1. Introduction

The red imported fire ant, *Solenopsis invicta* Buren (Hymenoptera: Formicidae), is a significant agricultural, ecological, and medical pest that has invaded many regions of the world [1,2]. The economic losses (including costs of damages and control efforts) caused by *S. invicta* are estimated at USD 6 billion in the United States each year [3]. Using toxic baits is one of the main methods to control *S. invicta* [4]. This method has some apparent advantages. First, the use of fire ant bait is flexible because it can be spread manually in small areas and applied on large scales using various types of spreading machinery, helicopters, aircraft, and unmanned aerial vehicles [5,6]. Second, bait treatment is cost-efficient (e.g., approximately USD 10–18 per acre) compared to other management methods, such as mound treatment (e.g., approximately USD 0.1–1 per mound) [7,8].

Many fire ant baits based on fipronil have been developed and applied for *S. invicta* control. For example, Allen and Miller [9] reported that the number of foraging fire ants decreased by 96%, 100%, 97%, 99%, 68%, and 9% after treating the plots with 0.0143% fipronil bait (Top Choice^®^, Bayer Environmental Science, Cary, NC, USA) for 90, 120, 240, 270, 300, and 360 d, respectively. Similarly, a laboratory-produced bait containing 0.01% fipronil had a control efficacy of 98% against *S. invicta* after 30 d of treatment in the field [10]. In addition, Yasudai et al. [11] reported that a paste-formulated bait (Hyper Arino-su-korori, Earth Corp., Tokyo, Japan) containing 0.002% fipronil killed almost all fire ants within 8 weeks in the laboratory and effectively decreased the number of foraging *S. invicta* in the field.

Some fire ant baits also use abamectin (a mixture of avermectin B1a and B1b) or indoxacarb as the active ingredient [12]. For example, Bhatkar [13] reported that plots treated with the Pt^®^ 370 Ascend^TM^ bait (Whitmire Research Laboratories, Inc., St. Louis, MO, USA) containing 0.011% avermectin B1 caused a significant decline in *S. invicta* activity compared to untreated plots within 30 d. Middleton et al. [14] reported that the application of Clinch fire ant bait (Syngenta Crop Protection. LLC, Greensboro, NC, USA) containing 0.011% abamectin successfully suppressed *S. invicta* populations in Florida citrus. Also, Oi and Oi [15] reported that no active *S. invicta* nests could be found in half of the treated areas three days after the application of Advion^®^ fire ant bait (Syngenta Crop Protection. LLC, Greensboro, NC, USA) containing 0.045% indoxacarb.

The effectiveness of a fire ant bait largely depends on its acceptance, or at least, non-repellency for *S. invicta* foragers. However, based on our best knowledge, very few studies have evaluated the behavioral responses of *S. invicta* to baits containing different active ingredients at different concentrations. Here, we hypothesized that the active ingredient in the fire ant bait would not significantly affect the bait transport behaviors of *S. invicta*. Field studies were conducted to evaluate the acceptance of fire ant bait containing different concentrations of each active ingredient (fipronil, abamectin, or indoxacarb) by comparing the mass of transported bait and the number of foraging ants with the control bait. Our previous study showed that *S. invicta* workers used soil particles to bury the food treated with repelling chemicals [16]. Therefore, we also measured and compared the weight of soil particles relocated in tubes containing toxicant-treated or control baits. In addition, laboratory studies were conducted to investigate the searching and transport behaviors of *S. invicta* workers in response to bait containing different active ingredients.

## 2. Materials and Methods

### 2.1. Bait Preparation

The method provided by Kafle and Shih [17] was modified to prepare fire ant baits. In brief, distiller’s dried grains with solubles (DDGS) were purchased from an online seller (Jiahui Feed Co., Ltd., Shijiazhuang, China). The DDGS was sifted through 2.0- and 0.9-mm sieves, and grains with particle sizes ranging from 0.9 and 2.0 mm were used as fire ant bait carriers. Fipronil (98%) and abamectin (97%) were purchased from Macklin Biochemical Technology Co., Ltd., Shanghai, China. Indoxacarb (97%) was purchased from Dr. Ehrenstorfer Co., Ltd., Ausberg, Germany. Fipronil and abamectin were dissolved in soybean oil (reagent grade, Macklin Biochemical Technology Co., Ltd., Shanghai, China) to reach a concentration of 20, 100, 500, or 2500 μg/g. Because the concentration of indoxacarb in the commercial fire ant bait is usually higher than fipronil and abamectin [9,13,14], we dissolved indoxacarb in the soybean oil to reach a concentration of 100, 500, 2500, or 12,500 μg/g. The soybean oil without toxicant was used as the control. The toxicant-treated or untreated oil and DDGS were then thoroughly mixed at a ratio of 1: 19 (*w*/*w*). Our preliminary experiment showed that the ratio of oil to DDGS (1:19) was satisfactory for ants to transport the bait, whereas the higher ratio gave rise to excessive oil content that could disturb the bait transport. The final concentration of fipronil and abamectin in the bait was 0% (control), 0.0001%, 0.0005%, 0.0025%, or 0.0125% (*w*/*w*), while the concentration of indoxacarb was 0%, 0.0005%, 0.0025%, 0.0125%, or 0.0625%.

### 2.2. Field Study

A certain amount of bait (2.95–3.05 g) was weighed using an electronic balance and transferred to the bottom of a 50 mL centrifuge tube (diameter = 28.5 cm, length = 116 mm, Labselected, Beijing, China). Our preliminary study showed that many *S. invicta* workers consumed the oil attached to the wall of the tubes instead of transporting the baits. Also, *S. invicta* workers could cover the surfaces treated with some pesticides [18]. To suppress these oil-consuming and particle-paving behaviors that may interfere with bait transport, we filled the void of tubes using cotton balls to prevent bait sloshing, and therefore oil would not be attached to the wall of the tubes during storage and transport before the experiment.

The field study was conducted at Zengcheng Teaching and Internship Base, South China Agricultural University (SCAU), Guangzhou, China. Massive *S. invicta* activities were previously detected in this area using sausage baits, and no pesticide treatment was applied >6 months before the experiment. The test for each toxicant was conducted separately. The fipronil bait was tested in a shrub zone (23.246° N, 113.632° E) on 15 October 2022; the abamectin bait was tested in a *Castanopsis hystrix* Miq. plantation (23.247° N, 113.631° E) on 17 October 2022; the indoxacarb bait was tested in a grassland (23.248° N, 113.632° E) >200 m apart from the *C. hystrix* plantation on 27 June 2023. The cotton ball was removed directly before the test, and the tubes were placed horizontally on the ground, with the entrance closely attached to the soil. Each concentration of each toxicant was repeated 15 times. In total, 75 tubes containing different concentrations of each toxicant were released with randomly assigned orders (each tube was >3 m apart). At the end of the experiment, all tubes were collected, sealed with lids, and brought to the laboratory. Ants were killed and stored in a −20 °C fridge.

The frozen tubes were placed at room temperature for 3 h to prevent the formation of condensed water due to the low temperature of the tube just taken out of the fridge. The contents of each tube were then poured out, and the bait grains, dead fire ants (no ant species other than *S. invicta* was found in the tubes in this experiment), and any soil particles were carefully separated using forceps and cardboard. The remaining bait grains were weighed using an electronic balance, and the mass of bait transported by *S. invicta* workers was determined by calculating the differences in bait weight before and after the experiment. The number of *S. invicta* workers in each tube was counted. In addition, soil particles found in each tube were weighed using an electronic balance.

### 2.3. Laboratory Study

Fifteen *S. invicta* colonies were collected and used in this study. Two colonies were collected from Zengcheng Teaching and Internship Base (23.244° N, 113.636° E), Guangzhou, China, on 11 July 2023, three colonies were collected from Furong Hill Park (23.510° N, 113.234° E), Guangzhou, on 22 July 2023, and the remaining colonies were collected from Maofeng Mountain (23.306° N, 113.445° E), Guangzhou, on 23 July 2023. Mound soil with eggs, larvae, pupae, and adults was rapidly shoveled into a plastic container—50 (L) × 37 (W) × 26 (H) cm—with walls previously smeared with the talcum powder to prevent *S. invicta* escape [19]. Collected ants were brought to the laboratory and extracted from the soil using the water-dropping method mentioned by Chen et al. [20]. For each colony, 5 g of workers and 0.5 g of immature ants (eggs, larvae, and pupae) were weighed and transferred to a plastic box—52 (L) × 35 (W) × 15 (H) cm—with walls coated with the talcum powder. A black cardboard paper (30 × 40 cm) was placed on the bottom of the plastic box, and an artificial nest (90 mm Petri dish containing dental plaster) was placed on the center of the cardboard (Figure 1). The lid of the artificial nest was covered by black cardboard to block light, and four entrances were drilled on the wall of the Petri dish. Ants were allowed to acclimatize for several days, and 20% honey water and frozen crickets were provided.

To encourage ants to transport the bait, food was no longer provided 24 h before the experiment. Directly before the experiment, 100 mg of control bait or bait containing fipronil (0.0025% or 0.0125%), abamectin (0.0025% or 0.0125%), or indoxacarb (0.0125% or 0.0625%) was weighed and placed on the center of a graph paper square (5 × 5 cm) coated with a plastic membrane. The seven squares were placed around the artificial nest in randomly assigned order. Each square was 4.5 cm apart from the edge of the artificial nest, and adjacent squares were equidistant. A 1 h video was taken for each arena. For each bait, we recorded the following: (1) duration for bait search (time between the beginning of the experiment and the first ant searching the bait with body contact); (2) duration for transport of the first bait particle (time between the first ant searching the bait and the first bait particle was transported away from the square); (3) total duration of bait transport (time between the first ant searching the bait and all bait particles were transported away from the square, or time between the first ant searching the bait and the end of the experiment if some bait remained untransported). In addition, the untransported bait on each square was collected and weighed at the end of the experiment, and the mass of transported bait was determined by calculating the differences in bait weight before and after the experiment. We then calculated the bait transport speed using the formula below:$$Bait\;transport\;speed=\frac{Mass\;of\;transported\;bait}{Total\;duration\;of\;bait\;transport}$$

### 2.4. Data Analyses

The normality of the data—mass of transported bait, number of ants in the baiting tube, weight of relocated particles for the field study, duration of bait search, duration of transport of the first bait particle, total duration of bait transport, mass of transported bait, and bait transport speed for the laboratory study—was checked using Proc Univariate (SAS 9.4, SAS Institute, Cary, NC, USA). If data were not normally distributed, log (x + 1) and square transformations were performed. If either the original or the transformed data were normally distributed, one-way analysis of variance (ANOVA) was conducted to compare data among treatments, followed by Tukey’s HSD tests for pairwise comparisons. Otherwise, the Kruskal–Wallis tests were used for data analyses, followed by Dwass–Steel–Critchlow–Fligner tests (DSCF) for pairwise comparisons. For all tests, *α* = 0.05.

## 3. Results

### 3.1. Field Study

*Solenopsis invicta* transported significantly less 0.0125% fipronil bait than control bait or 0.0001% fipronil bait, but they were not significantly different from 0.0005% or 0.0025% fipronil bait (Figure 2A). The number of foraging ants significantly decreased with the increase of fipronil concentration (Figure 2B). No ant was found in the tubes containing 0.0125% fipronil bait. Also, significantly more soil particles were transported into tubes containing 0.0001% fipronil bait than tubes containing control bait or 0.0125% fipronil bait (Figure 2C).

In addition, *S. invicta* transported significantly less 0.0005% abamectin bait than control bait, but they were not significantly different from the abamectin bait at other concentrations (Figure 3A). Significantly fewer ants were found in the tubes containing 0.0125% abamectin bait than controls (Figure 3B). There was no significant difference in weight of particles relocated into tubes containing different concentrations of abamectin bait (Figure 3C).

The mass of transported bait (Figure 4A), number of foraging ants (Figure 4B), and weight of soil particles (Figure 4C) relocated in tubes containing different concentrations (0%, 0.0005%, 0.0025%, 0.0125%, or 0.0625%) of indoxacarb bait was not significantly different.

### 3.2. Laboratory Study

The duration for bait search (Figure 5A), the duration for transport of the first bait particle (Figure 5B), and the total duration of bait transport (Figure 5C) were not significantly different among different baits. However, significantly less 0.0125% fipronil bait was transported than for other baits tested (Figure 5D). Also, the transport speed of 0.0125% fipronil bait was significantly slower than other bait (Figure 5E).

## 4. Discussion

Because of the success of commercial fipronil-based baits against *S. invicta*, it appeared that fipronil bait would be non-repellent to fire ants at the applied concentrations. Hooper-Bùi et al. [21] reported that *S. invicta* actively visited and transported the MaxForce bait (The Clorox Service Company, Pleasanton, CA, USA) containing 0.001% fipronil. Similarly, our field study showed that *S. invicta workers* transported a similar mass of control bait and fipronil bait at concentrations equal to or less than 0.0025% (Figure 2A). However, significantly less 0.0125% fipronil bait was transported by ants compared with the control bait in both field and laboratory studies, indicating the repellency of 0.0125% fipronil bait against *S. invicta*. In our laboratory study, 0.0125% fipronil did not significantly increase the duration of bait search and transport of the first bait particle (Figure 5A,B), indicating that 0.0125% fipronil may not negatively affect the initial stage of bait transport. However, 0.0125% fipronil significantly decreased bait transport speed (Figure 5E), probably because fewer ants were involved with bait foraging, as shown in the field study (Figure 2B). McCreery and Breed [22] provided a method to measure the food transport efficiency, which is equal to the mass of food times the velocity vector, divided by the number of transporters. However, in our study it was challenging to separate the ants that stayed on or around the bait to consume the oil and the ants that directly transported bait particles.

It is worth noting that some commercial fire ant baits may contain higher concentrations of fipronil (e.g., the fipronil concentration in the Top Choice^®^ bait is 0.0143%). However, our study does not contradict previous studies showing the effectiveness of these fire ant baits. First, some baits may use various attractants that could counteract the repellency of the active ingredient. For example, Kafle and Shih [17] developed a fire ant bait using shrimp shell powder as the lure, which could cover the repelling effect of cypermethrin. Second, it might be sufficient to suppress *S. invicta* populations even if only a small portion of the broadcast bait were to be foraged by *S. invicta*. Nevertheless, decreasing the fipronil concentration may increase the bait acceptance by fire ants, and therefore induce the control efficiency of the fipronil bait. It is also important to note that too low concentrations of fipronil may not be sufficient to kill fire ants. Therefore, field studies are needed to determine the range of non-repellent fipronil concentrations that can effectively eliminate fire ants.

Interestingly, recent studies showed that fipronil can affect many behaviors and physiological processes of insects. For example, Rosa et al. [23] reported that sublethal concentrations of fipronil significantly induced grooming behavior but decreased exploratory activity, learning, and olfactory memory of the cockroach *Nauphoeta cinerea* (Olivier). Likewise, fipronil may have various effects on *S. invicta*. For example, Chen and Allen [24] reported that *S. invicta* excavated fipronil-treated sand (1 and 10 ppm) in the two-choice tests when the untreated sand was also provided, indicating that fipronil sand did not repel digging ants. However, Wen et al. [18] reported that the surface treatment of 500 or 5000 μg/mL fipronil solution significantly decreased the number of *S. invicta* workers gathered in a sausage lure placed on the treated surfaces compared to the control ones.

Baker et al. [25] reported that avermectin added in sucrose water did not deter the feeding behavior of the Argentine ant, *Iridomyrmex humilis* (Mayr), compared with the control of sucrose water. However, Kaul et al. [26] reported that peanut butter containing abamectin was ignored by *S. invicta* and the longhorn crazy ants (*Paratrechina longicornis* (Latreille)), which quickly recoiled after a few ants approached and tested the abamectin-treated food with their antennae. In our study, *S. invicta* workers transported less 0.0005% abamectin bait than the control bait (Figure 3A). In addition, fewer ants were found in the tubes containing 0.0125% abamectin bait than the control bait (Figure 3B). These results show that the abamectin bait may negatively affect the foraging behavior of *S. invicta.* However, it is worth noting that ants transported a similar amount of 0.0125% abamectin bait and control bait in the field and laboratory. Therefore, the concentration of abamectin in commercial baits such as Pt^®^ 370 Ascend^TM^ (containing 0.011% avermectin B1) and Clinch (containing 0.011% abamectin) may not negatively affect the bait transport in *S. invicta*.

Some recent studies focused on the mechanism and novel formulations of indoxacarb against *S. invicta.* For example, Du et al. [27] applied mass spectrometry imaging and untargeted metabolomics technologies to reveal the disturbance of several key metabolic pathways in different tissues of *S. invicta* bodies after indoxacarb exposure. Siddiqui et al. [28] reported the changes in the transcriptome profile and detoxification enzyme activities of *S. invicta* treated with sublethal concentrations of indoxacarb. Yang et al. [29] developed a smart pH and α-amylase dual stimuli-responsive pesticide delivery system that can reduce the photodegradation and increase the toxicity of indoxacarb against *S. invicta.* Our study showed that the indoxacarb bait is non-repellent to fire ants at the concentration of 0.0625%, which is higher than that used in the commercial Advion^®^ fire ant bait (0.045% indoxacarb). However, some indoxacarb baits applied in China have a concentration of 0.1% active ingredient [6]. It would be valuable to evaluate the foraging behaviors of *S. invicta* workers in response to the bait containing higher concentrations of indoxacarb.

Fire ants usually show many object-use behaviors [30]. For example, our previous studies show that *S. invicta* usually relocated soil particles to cover the surfaces treated with pesticides, repellents, or sticky materials [18,31,32]. In addition, we observed that *S. invicta* workers used particles to bury the food under satiation conditions or when the food was treated with repelling chemicals [16]. Both behaviors are involved with particle utilization and are associated with the foraging processes of *S. invicta*. However, the paving behavior facilitates food search and transport [31], whereas the burying behavior occurs when ants are unwilling to forage for food [16]. In this study, procedures were performed to avoid attaching the oil to the wall of the tubes, which may create sticky and pesticide-treated surfaces that may trigger paving behavior. However, we still observed some soil particles relocated into the tubes, and more soil particles were found in the tubes containing 0.0001% fipronil bait than control bait or 0.0125% fipronil bait (Figure 2C). We believe this was caused by the burying behavior of ants responding to specific concentrations of fipronil. Interestingly, the weight of relocated soil particles did not continue to increase with higher fipronil concentrations, probably due to the suppression of foraging-related behaviors, including soil relocation at the higher concentrations. Since indoxacarb did not repel foraging ants, we observed no significant difference in the weight of soil particles among baiting tubes containing different concentrations of indoxacarb. It would be interesting to investigate whether *S. invicta* would bury the broadcast baits containing fipronil or other active ingredients.

One possible drawback of our study is that we only conducted the field study for each active ingredient once. However, choosing the locations for our field studies is challenging, where *S. invicta* should be the dominant ant species foraging the oil-based bait. Otherwise, other ant species may enter the baiting tubes and transport bait particles. In our preliminary experiments, some ant species may have transported baits and then left, and we cannot be sure whether *S. invicta* transported the bait. Inclusion of these baiting tubes may have caused errors in evaluating the bait transport of *S. invicta*. Our laboratory study is consistent with the field study, indicating that our results are valid and reproducible. We suggest conducting laboratory and field tests to evaluate the acceptance of newly-developed fire ant baits in future studies.

## 5. Conclusions

Our study shows that 0.0125% fipronil bait is repellent against *S. invicta* workers. Although the 0.0125% abamectin bait decreased the number of foraging ants in the field, it did not negatively affect bait transport. In addition, indoxacarb had no significant effect on bait transport of fire ants at the tested concentrations, indicating that indoxacarb bait is non-repellent to *S. invicta*. In future studies evaluating the effectiveness of fire ant bait, we suggest considering the effect of active ingredients on bait acceptance.

## Figures and Tables

**Figure 1 insects-14-00852-f001:**
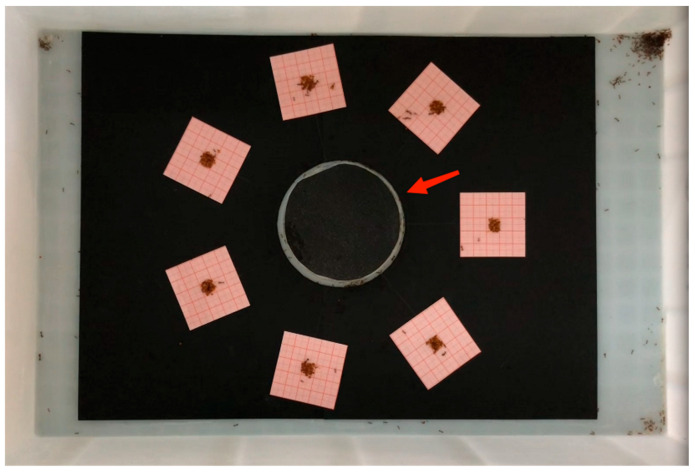
Bioassay arena of laboratory experiment to investigate the search and transport behaviors of *Solenopsis invicta* workers in response to control bait or bait containing different concentrations of fipronil (0.0025% or 0.0125%), abamectin (0.0025% or 0.0125%), or indoxacarb (0.0125% or 0.0625%). The red arrow indicates the location of the artificial nest.

**Figure 2 insects-14-00852-f002:**
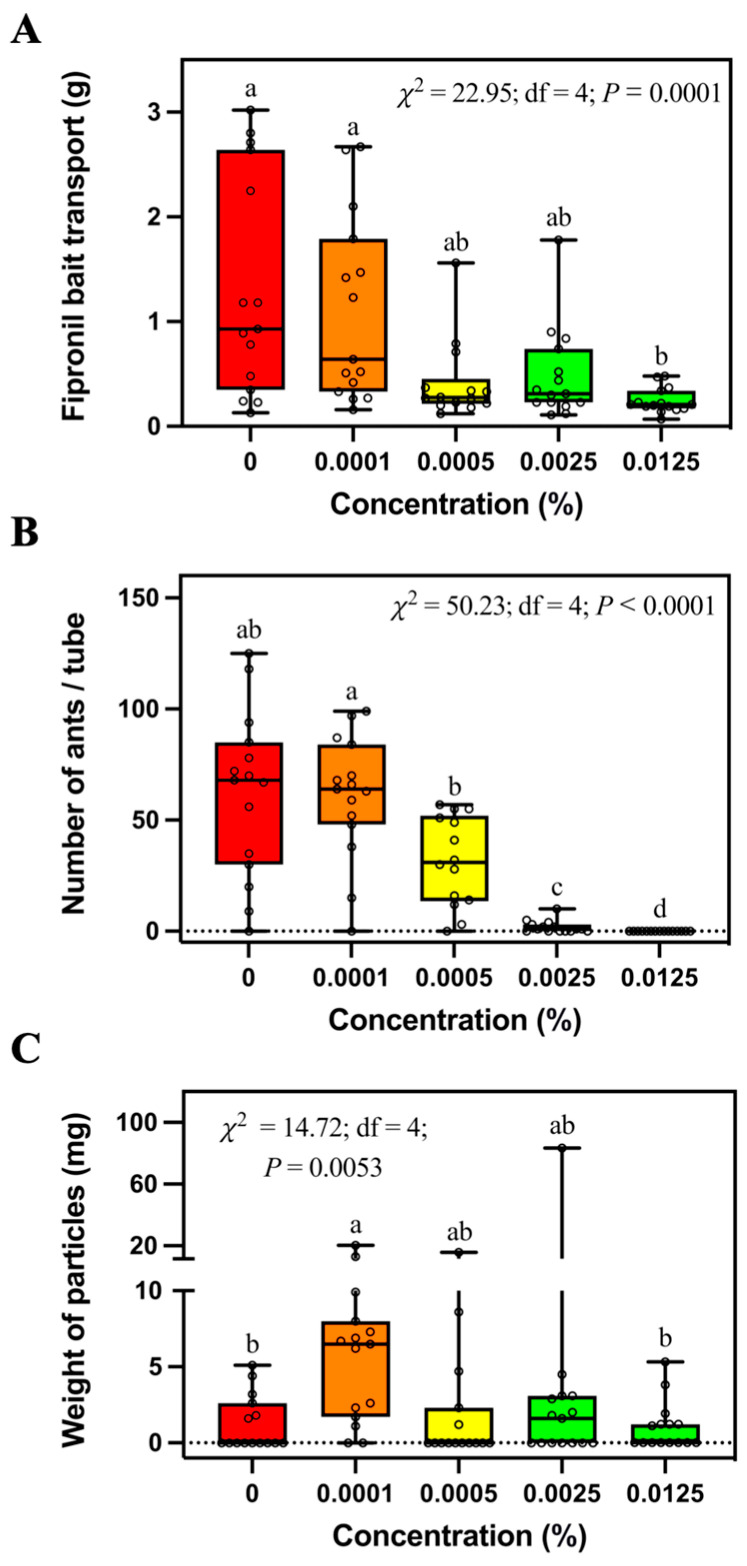
Mass of transported bait (**A**), number of foraging ants (**B**), and relocated soil particles (**C**) in the tubes containing different concentrations of fipronil bait. Boxes show the 25th percentile, 50th percentile (median), 75th percentile; whiskers show the maximum and minimum value of the data; hollow spots show the value of each data. Different letters indicate significant differences (pairwise comparisons were conducted by Tukey’s HSD tests for normally distributed data or Dwass–Steel–Critchlow–Fligner tests for non-normally distributed data, *p* < 0.05).

**Figure 3 insects-14-00852-f003:**
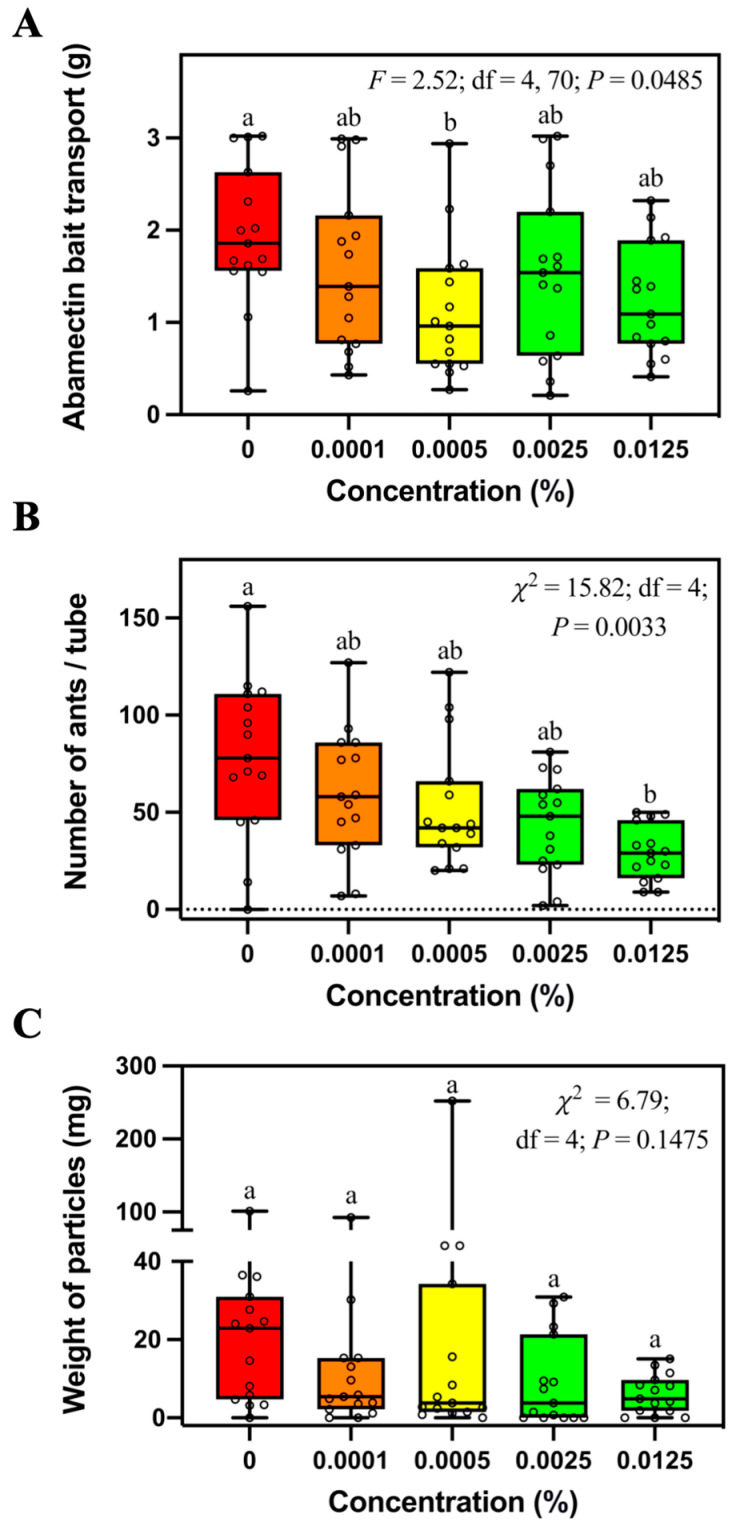
Mass of transported bait (**A**), number of foraging ants (**B**), and relocated soil particles (**C**) in the tubes containing different concentrations of abamectin bait. Boxes show the 25th percentile, 50th percentile (median), 75th percentile; whiskers show the maximum and minimum value of the data; hollow spots show the value of each data. Different letters indicate significant differences (pairwise comparisons were conducted by Tukey’s HSD tests for normally distributed data or Dwass–Steel–Critchlow–Fligner tests for non-normally distributed data, *p* < 0.05).

**Figure 4 insects-14-00852-f004:**
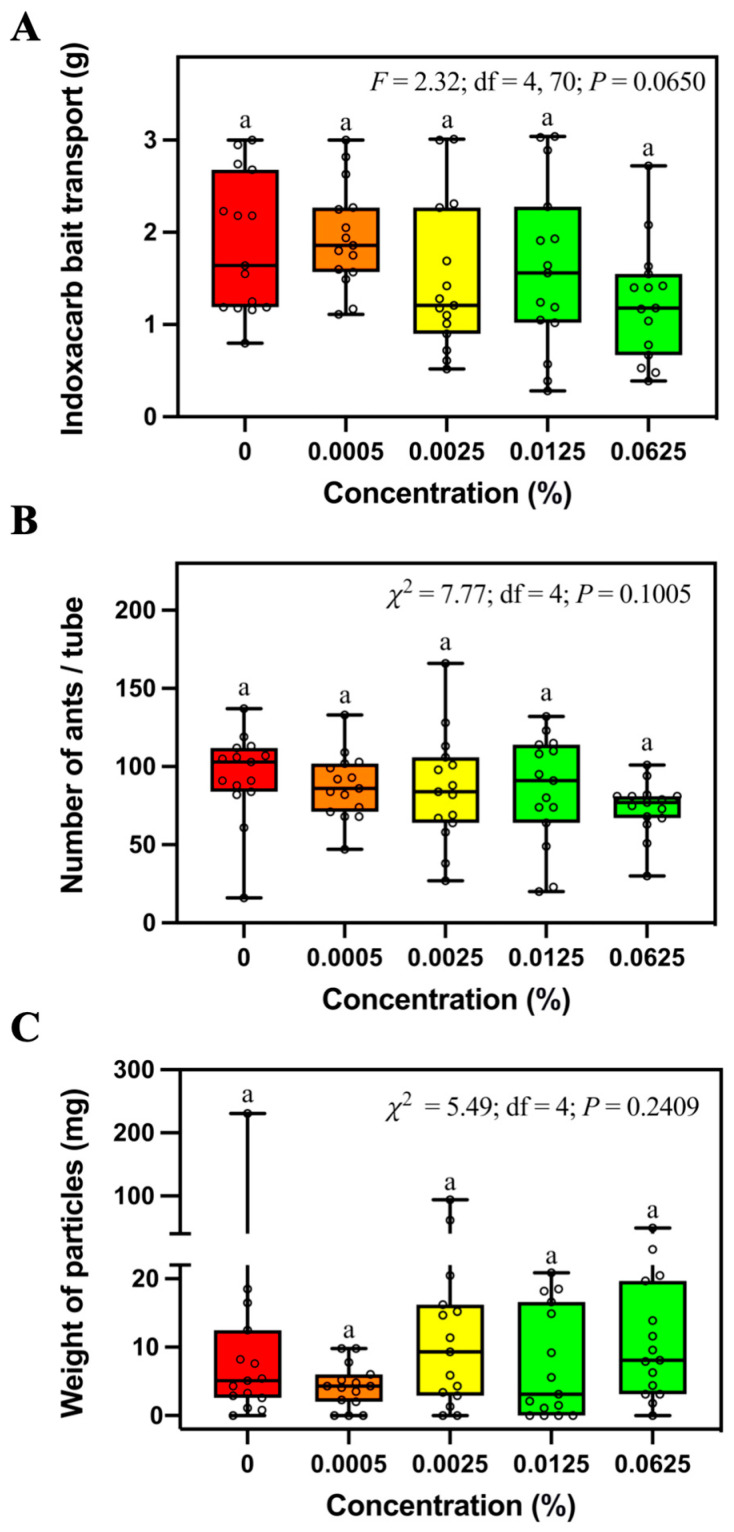
Mass of transported bait (**A**), number of foraging ants (**B**), and relocated soil particles (**C**) in the tubes containing different concentrations of indoxacarb bait. Boxes show the 25th percentile, 50th percentile (median), 75th percentile; whiskers show the maximum and minimum value of the data; hollow spots show the value of each data. Different letters indicate significant differences (pairwise comparisons were conducted by Tukey’s HSD tests for normally distributed data or Dwass–Steel–Critchlow–Fligner tests for non-normally distributed data, *p* < 0.05).

**Figure 5 insects-14-00852-f005:**
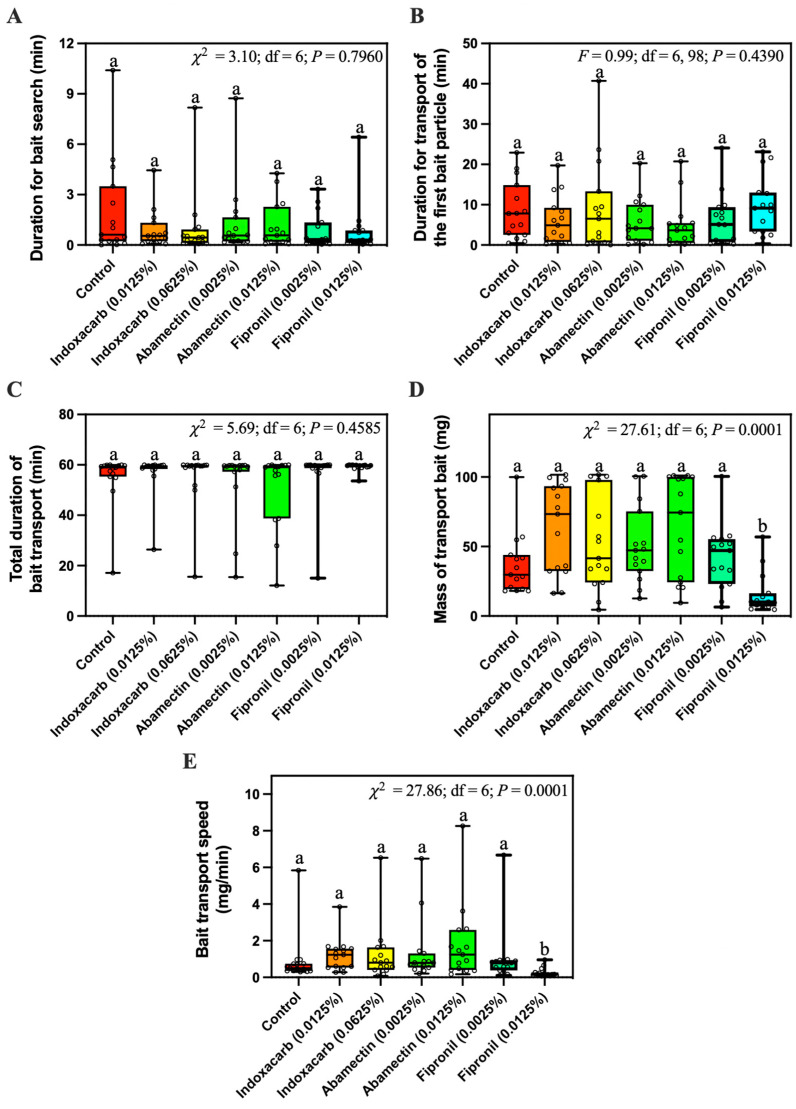
Duration for bait search (**A**), duration for transport of the first bait particle (**B**), total duration of bait transport (**C**), mass of transported bait (**D**), and bait transport speed (**E**) of *Solenopsis invicta* workers in response to control bait or bait containing different concentrations of fipronil (0.0025% or 0.0125%), abamectin (0.0025% or 0.0125%), or indoxacarb (0.0125% or 0.0625%). Boxes show the 25th percentile, 50th percentile (median), 75th percentile; whiskers show the maximum and minimum value of the data; hollow spots show the value of each data. Different letters indicate significant differences (pairwise comparisons were conducted by Tukey’s HSD tests for normally distributed data or Dwass–Steel–Critchlow–Fligner tests for non-normally distributed data, *p* < 0.05).

## Data Availability

The raw data and materials will be made available by the authors, without undue reservation, to any qualified researchers.

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
