# Peer review of "Foraging Behaviors of Red Imported Fire Ants (Hymenoptera Formicidae) in Response to Bait Containing Different Concentrations of Fipronil, Abamectin, or Indoxacarb"

_insects, 2023, doi:10.3390/insects14110852_

Round 1

Reviewer 1 Report

Comments and Suggestions for Authors

The manuscript titled "Foraging Behaviors of Red Imported Fire Ants (Hymenoptera: Formicidae) in Response to Bait Containing Different Concentrations of Active Ingredient (No. insects-2636923)" investigates the transport behaviors of red fire ants in both field and indoor laboratory studies when exposed to oil-based baits with varying concentrations of fipronil, abamectin, or indoxacarb. This study makes a valuable contribution to the development of new, more effective, non-repellent red fire ant baits. The topic and content are suitable for publication in Insects and its special issue, with minor modifications as outlined below:

[1]      Page 1, Line 28-29: Please provide additional data and results from the laboratory study.

[2]      Page 1, Line 30: Please explain how this study can advance bait technology and improve its efficacy.

[3]      Page 2, Line 69: State the hypothesis that guided this study for clarity and to ensure readers clearly understand the research objectives.

[4]      Page 2, Line 90-91, Was the ratio of oil to DDGS (1:19) obtained by screening.

[5]      Page 3, Line 122-124: In field trials, please clarify whether a control group was established to account for variations in bait weight caused by environmental factors (e.g., humidity, temperature, etc.).

[6]      Page 4, Line 159: Did the same ant search the bait and transported the particles away from the square?

[7]      The manuscript still contains some formatting and grammatical errors; please correct them. For example, Line 16, Line 181, Line 325.

[8]      In Figure 1, consider adding labels or arrows to indicate the location of the artificial nest to enhance clarity for readers.

Reviewer 2 Report

Comments and Suggestions for Authors

See file for reviewer comments

Reviewer 3 Report

Comments and Suggestions for Authors

I have carried out a review of the article “Foraging Behaviors of Red Imported Fire Ants (Hymenoptera Formicidae) in Response to Bait Containing Different Concentrations of Active Ingredient.” by Du et al.

I attest the study was well conducted and quite interesting. In addition, the manuscript was well written and the organization is of good standard. The experimental procedures were also clearly highlighted. I do not have any reservations about the acceptance of the manuscript for publication in the insect journal.

However, I have a few suggestions that could be effected to improve the manuscript.

L11 – Revise as “studies show that”

L12 – Delete “and” Revise as “…workers; therefore,

L12 – Delete “the” Revise as “…therefore, high concentrations of fipronil…”

L12 – Revise as “should be avoided”

L13 – Revise as “fire ant baits”

L22 – Revise as “with an increase in fipronil concentration”

L85 – Revise as “…were dissolved in soybean oil…”

L267 - Revise as “…some commercial fire ant baits…”

L269 - Revise as “…these fire ant baits”

Comments on the Quality of English Language

The English quality/writing is relatively fine and of an acceptable standard. Authors are required to make a few minor corrections to grammar and expressions. I have highlighted a few corrections that should be made.

Reviewer 4 Report

Comments and Suggestions for Authors

This manuscript assessed the attractiveness of three different active ingredients at different concentrations to red imported fire ants by assessing how much bait was removed, how many ants were present, and the timing of these events. Additionally, they assessed how much dirt was placed on the baits by the ants. They did this by conducted a field study comparing multiple concentrations per active ingredient and with a laboratory study comparing the active ingredients represented by two concentrations each. The authors found that higher concentrations of fipronil were generally taken less by the ants relative to lower concentrations but that higher concentrations of the other two active ingredients did not appear to alter ant repellency.

This study is an important contribution to assessing mechanisms of effective baits by evaluating how attractive difference concentrations and active ingredients are. The discussion of the findings could generally be improved by putting these results into the context of what the lowest known lethal concentrations and by discussing the soil packing a bit more. Otherwise, the manuscript was well written.

Line by Line comments

Title: Specify three active ingredients.

Simple summary, Line 12: “avoid” should be “avoided”.

Simple summary, Line 14: Specify “active ingredient concentration” here.

Abstract, Line 17: Change to “…few studies have evaluated…”

Abstract, Lines 22-23: Add in some information on the context for measuring the soil particles here.

Introduction, Line 34: Specify the order and family of the red imported fire ant.

Introduction, Line 45: Recommend adding in the costs of surface and mound treatments for ease of comparison here.

Introduction, Lines 66-76: Add in/move the context for measuring the soil particles to this paragraph. I would also recommend more explicitly specifying in this paragraph that the field study compared concentrations per active ingredient while the laboratory study compared 2 concentrations of each AI for all AIs.

Materials and Methods, Line 81: Add the seller information for the DDGS.

Materials and Methods, Lines 99 and 101: Replace “sucked” and “sucking” with “consumed” and “consumption” or similar terms. Same with the use of “suck” in the Discussion, Line 266.

Materials and Methods, Line 157: Update with “…the beginning of the experiment…”

Materials and Methods, Line 173: Change “are” to “were”.

Figures: Specify the active ingredient being assessed in the figure legend of each panel grouping.

Results, Lines 185-187: It might be worth noting somewhere in the discussion that less dirt was found at the higher concentrations of fipronil since ants were appearing to avoid those treatments more. Otherwise, you might expect that dirt weight would just keep increasing with higher concentrations.

Similarly, it would be good to add to the discussion that the AI differences in ant repellency may account for the lack of significant differences in dirt weight with abamectin and indoxacarb.

Discussion: Insert a new paragraph at the beginning of the discussion giving a quick summary of the project goals and overall findings to help the context up again.

Discussion, Lines 240-249: This paragraph (on how repellency might happen) would flow better after the following paragraph that sets up the context that fipronil baits can be repellent.

Discussion, Line 250: Adding in information about the lowest known lethal dose of fipronil would be useful here.

Discussion, Line 270: Change “bait” to “baits”.

Comments on the Quality of English Language

A few minor grammatic and phrasing issues were noted in the line by line comments, but generally it was well written.
